# Effects of plain water intake before bedtime on sleep and depressive mood among middle-aged Japanese men

Kosuke Kaida[1]*, Kazane Itoi[1], Yoshiharu Soeta[2], Toshihiko Ooie[3,4], Katsutaka Oishi[3,5,6]

1 Institute for Information Technology and Human Factors, National Institute of Advanced Industrial Science and Technology (AIST), Tsukuba, Ibaraki, Japan, 2 Molecular Biosystems Research Institute, National Institute of Advanced Industrial Science and Technology (AIST), Ikeda, Osaka, Japan, 3 Cellular and Molecular Biotechnology Research Institute, National Institute of Advanced Industrial Science and Technology (AIST), Tsukuba, Ibaraki, Japan, 4 Medilux Research Center, Nara Institute of Science and Technology (NAIST), Ikoma, Nara, Japan, 5 Department of Applied Biological Science, Graduate School of Science and Technology, Tokyo University of Science, Noda, Chiba, Japan, 6 Department of Computational Biology and Medical Sciences, Graduate School of Frontier Sciences, The University of Tokyo, Kashiwa, Chiba, Japan

* kaida.kosuke@aist.go.jp

## Abstract

Although adequate water intake during the day enhances mental health, its effects during sleep remain unknown. We aimed to define the impact of plain water intake before bedtime on sleep parameters and depressive mood. We surveyed 2,000 Japanese using a questionnaire to determine associations between pre-bedtime water intake and depressive mood measured using the Center for Epidemiologic Studies Depression Scale (CES-D; Study 1). We also compared the effects of consuming 280 mL of plain water immediately before bedtime and of not consuming any beverages for at least 2 hours before bedtime (Study 2). The findings of Study 2 indicated that plain water intake immediately before bedtime decreased depressive mood in the morning, prolonged REM sleep latency, and reduced the duration of REM sleep. However, plain water intake also increased the likelihood of nocturia. Although balancing advantages and disadvantages is essential, the practice of consuming plain water before bedtime is a simple and effective way to enhance subjective wellbeing.

## Introduction

Water accounts for 50%–70% of human body weight [1]. Water is crucial for survival and a critical physiological issue in terms of hypohydration, or dehydration [2]. The physiological water content is associated with blood pressure regulation [3] and all animals require water to rehydrate and alleviate thirst [4]. Adult humans should drink at least 2 L of plain water daily to prevent dehydration and maintain hydration.

Water constantly evaporates from the surface of the body not only while awake, but also during sleep [5]. This means that 400–700 mL of water evaporate from

---

**Data availability statement:** All relevant data are within the manuscript and its Supporting Information files.

**Funding:** The study was funded by Suntory Beverage & Food Limited (Tokyo, Japan).

**Competing interests:** The authors have declared that no competing interests exist.

the body while sleeping during the night. Because oral fluid intake during sleep is impossible by nature, water intake before bedtime should help to prevent nighttime dehydration.

Daytime dehydration is positively associated with irritation, anger, and depressive mood [2,6–10]. Conversely, the optimal daily consumption of plain water (euhydration) is inversely associated with depressive mood and suicidal thoughts [11,12]. Because symptoms of dehydration can develop throughout the day, water consumption at night also seems important. We speculated that water intake before bedtime would improve mood in the morning, and if so, avoiding dehydration during the night should improve subjective wellbeing [1,2,13].

The present study targeted the relationship between dehydration and depressive mood, which might be a leading cause of poor subjective wellbeing. The global estimated ratio of adults with depression is 10.8%, of which 1.1% is severe [14], and 3%–7% of the Japanese population has experienced severe depression at least once during their lifetime [15]. The main symptoms of depression are persistently low mood, difficulty concentrating, slowed mental activity, cognitive impairment, thoughts about death including suicide, and disrupted sleep [16]. The ratio of mortality caused by suicide among the Japanese population is presently > 16 per 100,000, which is higher than that of other G7 nations such as the USA, UK, and Canada [17]. Therefore, understanding associations between hydration during sleep, depressive mood, and other related factors is important to enhance subjective wellbeing, particularly in countries with alarming depressive trends.

Daytime hydration and depressive mood are negatively associated. However, the relationship between nighttime hydration and depressive mood remains unknown. The present study aimed to clarify the effects of hydration with plain water on sleep parameters and depressive mood in the morning. We determined the relationship between thirst (hypohydration), sleep quality (insomnia) and depressive mood using an online questionnaire survey (Study 1). We also assessed the effects of plain water intake on physiological sleep parameters at night and depressed mood upon waking in a cohort of healthy middle-aged Japanese males (Study 2). These studies are part of a larger, privately-funded project that aims to determine the effects of hydration on human physical and psychological states.

Study 1: We collected pilot information about relationships between morning thirst and sleep satisfaction (insomnia) as well as depressed mood. Although water intake is considered as an effective approach to ameliorate depressive mood, it also has negative associations [7,11,12,18–21]. Water intake is associated with low depressive mood and fewer suicidal behaviors in Korea, which has a high suicidal ratio [17]. However, most studies have explored the effects of daily water intake. Therefore, the effects of water consumption at night have remained unknown as far as we can ascertain. We posited that respondents who frequently consumed water before bedtime would experience decreased depressive mood in the morning as well as less insomnia that would indicate improved sleep satisfaction.

Study 2: We compared conditions with and without plain water intake before bedtime to establish causal relationships between these factors. Sleep parameters were

calculated from polysomnographic data. Depressive mood has been associated with rapid eye movement (REM) and non-REM (NREM) sleep [22–24]. Among the parameters associated with REM sleep, latency (elapsed time from onset of first NREM sleep to that of the first REM sleep) decreases and the amount of REM sleep increases among patients with depression. Among the parameters associated with NREM sleep, decreased slow wave sleep (SWS), increased sleep latency and frequent wakening during the night are associated with depression [22–24].

We assumed that plain water intake before bedtime would increase REM sleep latency and decrease the amount of REM sleep, which could ameliorate depressive mood and morning thirst. Our hypotheses regarding water intake at night just before going to sleep were that plain water intake decreases depressive mood and thirst, increases REM sleep latency, decreases the amount of REM sleep, increases SWS, and decreases sleep latency as well as the frequency of awakening.

## Study 1

### Methods

**Participants and survey.** We selected 2,000 Japanese aged ≥ 18 years from a large online panel (Rakuten Insight, Inc., Tokyo, Japan) using stratified random sampling based on the age and sex distribution in all Japanese prefectures. Invitations to participate in the survey were e-mailed. The ethnic identity of all respondents was Asian, and all were native Japanese speakers. Participants who completed the survey received a token to access diverse e-commerce services provided by the Rakuten Group, which is the largest platform in Japan.

Participants were given a questionnaire (as part of a larger research survey) comprising 119 and 120 questions for males and female, respectively, that required ~ 20 minutes to answer. We analyzed morning thirst, water intake before bedtime, depressive mood (determined by the Center for Epidemiologic Studies Depression Scale, CES-D; 20 items [25]), and insomnia (determined by the Insomnia Severity Index, ISI; 7 items [26]) in Study 1. The participants rated morning thirst after awakening from sleep using a 7-point Likert scale ranging from "*not at all*" to "*extremely thirsty.*" Water intake before bedtime was assessed using a 7-point Likert scale ranging from "*strongly disagree*" to "*strongly agree.*"

Respondents with poor health and under medication at the time of the survey were excluded, resulting in 995 valid responses from 469 males and 526 females aged 18–29, 30s, 40s, 50s, 60s, 70s, + 80s (n = 234, 212, 162, 159, 112, 104, and 12, respectively).

**Compliance with ethical standards.** The Ethics Review Board at the National Institute of Advanced Industrial Science and Technology approved the study (Approval ID: 73002030-E-20231201–001), which complied with the ethical principles enshrined in the Declaration of Helsinki (2013 amendment). All respondents were recruited during the 20th–29th of December 2023 provided written, informed consent to participate.

**Data analysis.** The summed scores of CES-D and ISI and the thirst score were classified into seven groups based on seven water intake scores (e.g., Group 1: thirst score 1). Linear relationships between water intake and thirst were analyzed using one-way ANOVA and post-hoc Tukey HSD tests. The strength of linear relationships scored between two scales (variables) was calculated using Pearson correlation coefficients. All data were analyzed using IBM SPSS version 29 for Windows.

### Results

**Study 1.** S1 Table shows all the statistical findings.

**Depressive mood.** The one-way ANOVA detected a significant effect of water intake ($F$ [6, 994] = 7.87, $p < 0.01$, $\eta^2 = 0.05$). Post-hoc analysis found higher CES-D scores in group 3, than in groups 5–7 ($p < 0.05$; Fig 1A). The CES-D scores revealed an inverted U-shaped trend for water intake. The correlation coefficient between water intake and CES-D scores was $r = −0.18$ ($p < 0.01$), indicating that less water intake was associated with higher CES-D scores.

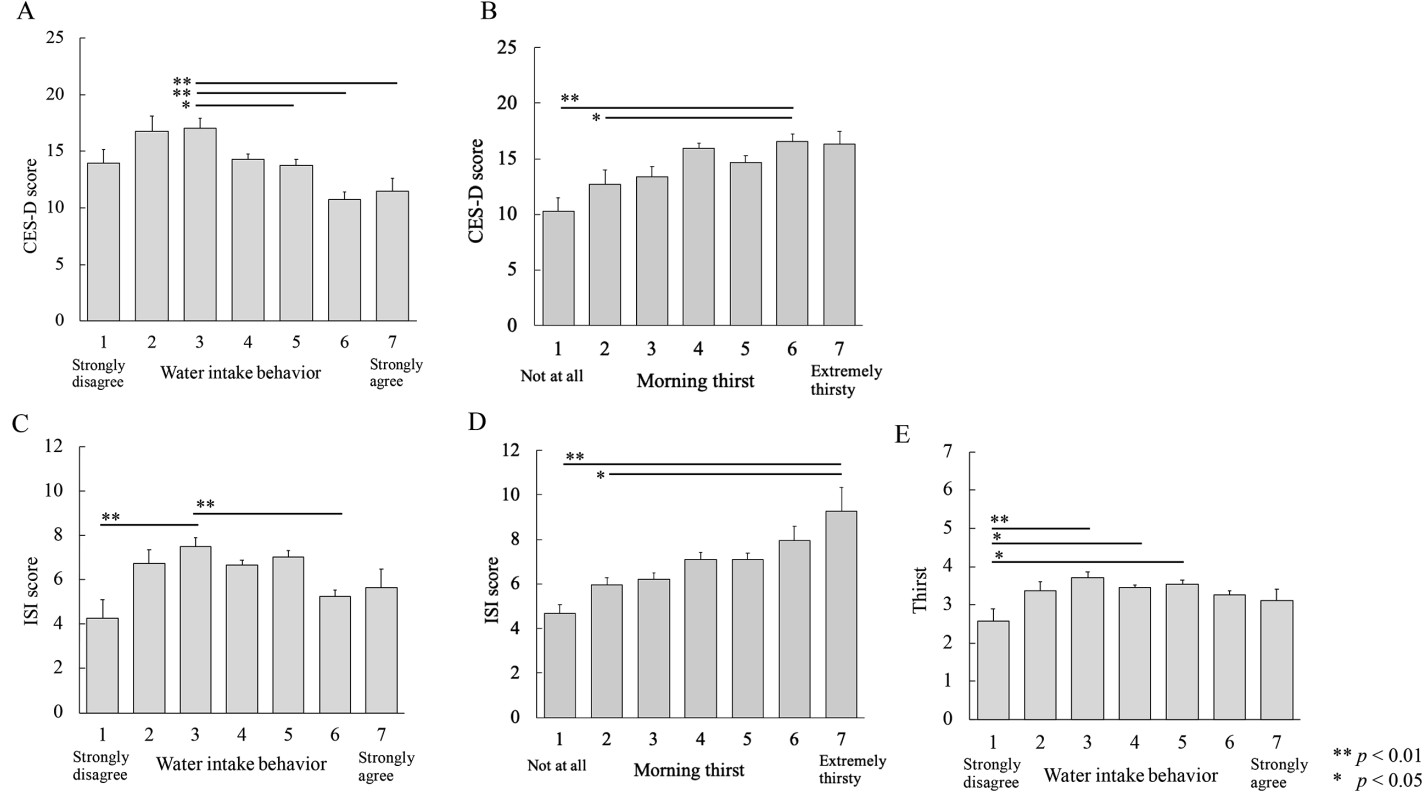

**Fig 1. Relationships among depression, insomnia, water intake behavior and thirst in the morning.** (A) Lower water intake was associated with higher CES-D scores. (B) Depressive mood associated positively with morning thirst. (C) Relationship between insomnia and water intake was an inverted U. (D) More thirst was associated with worse insomnia scores. (E) Relationship between water intake and thirst was also an inverted U. CES-D: Center for Epidemiologic Studies Depression Scale; ISI: insomnia severity scale.

One-way ANOVA detected a significant effect of thirst ($F$ [6, 994] = 7.30, $p < 0.01$, $\eta^2 = 0.04$). The CES-D and thirst scores were typically higher in group 6 than in groups 1 and 2 ($p < 0.05$; Fig 1B). The relationship between these variables was essentially linear. The correlation coefficient between thirst and CES-D scores was $r = 0.18$ ($p < 0.01$) indicating that thirst correlated with the severity of depressive mood. Therefore, depressive mood associated negatively and positively with water intake and thirst.

**Insomnia.** One-way ANOVA identified a significant effect of water intake ($F$ [6, 994] = 5.46, $p < 0.01$, $\eta^2 = 0.03$). The ISI (insomnia) score was higher in group 3 than in groups 1 and 6 ($p < 0.01$), and the relationship with water intake was an inverted U (Fig 1C). The correlation coefficient between water intake and ISI scores was $r = -0.06$ ($p = 0.07$).

One-way ANOVA also revealed a significant effect of thirst ([6, 994] = 7.37, $p < 0.01$, $\eta^2 = 0.04$). The ISI score was the highest in group 7 compared with groups 1 and 2 ($p < 0.05$). The correlation coefficient between thirst and ISI scores was $r = 0.20$ ($p < 0.01$). This means that more thirst was associated with worse insomnia scores (Fig 1D).

**Morning thirst.** One-way ANOVA identified a significant effect of water intake ($F$ [6, 994] = 3.26, $p < 0.01$, $\eta^2 = 0.02$). Post-hoc analysis revealed lower thirst scores were in group 1 than in groups 3–5 ($p < 0.05$; Fig 1E), whereas those of groups 1, 6, and 7 did not significantly differ. The relationship between water intake and thirst was an inverted U. The correlation coefficient between water intake and thirst scores was $r = 0.01$ ($p = 0.78$).

## Study 2

### Participants and experimental design

This study included 63 healthy Japanese volunteers (age at the time of consent, 50–59 y) who met the following inclusion criteria between May and August in 2024 (spring and summer seasons in Japan): males, worked at a desk during weekdays, body mass index (BMI) 18.5–25.0, moderate type assessed by a Morningness/Eveningness Questionnaire, did not regularly drink water just before sleep, had no sleep disorders such as apnea, lifestyle-related illnesses, kidney issues, urinary organ disorders, and not consume food supplements or medications; did not regularly exercise after 5 pm, did not smoke within 2 hours before sleep, or consume alcoholic beverages, coffee or tea within 2 hours before going to sleep, and had not traveled within the past month to destinations that differed by ≥ 2 time zones. The average BMI was (22.7 ± 2.14 kg/m²). We confirmed that the circadian rhythms self-reported by the participants were all within a moderate range in terms of the above indicators and the average Morning/Eveningness score [27,28] was 53.1 ± 3.20.

Eight participants were excluded from analysis as outliers because of being unable to complete the entire experimental schedule (n = 5), insufficient sleep for at least one night during the experiment (n = 1), and two did not provide polysomnography data due to technical errors. The data of 55 male participants (mean age 54.6 years, SD = 2.90) were analyzed using a within-comparison design as follows: control (no water intake 1 hour before bedtime for 6 days from the first day of the experiment), water intake (280 mL plain water immediately before bedtime for 6 days before and on the day of the experiment). The participants adhered to the same daily routine at home according to whether they were assigned to consume water before bedtime for 6 days before the experiment day. A seven-day wash-out period (no water intake restrictions before bedtime) was inserted between the two experimental schedules (Fig 2). The time frame of the entire experiment including a wash-out period was 3 weeks. The participants noted their water and beverage consumption during the experimental period, and we counterbalanced the order of conditions among them.

### Experimental schedule

Aside from the daily routine at home with or without water intake at night, the experiment proceeded in a temperature-controlled private hotel room. The participants were acclimated to the experimental environment by spending the night before the experiment in the hotel. On the day before starting on Day 1 (Fig 2), those in the experimental group provided written, informed consent, responded to health questionnaires, practiced the psychomotor vigilance test (PVT), then were given bottles of plain water for 6 days before bedtime at home.

Fig 3 shows the standard experimental schedule. Polysomnographic data were recorded from participants on the experimental days using electrodes attached to their faces. The experiment started at 18:00. All participants showered, had dinner, then had free time until 23:00, when they were prohibited from consuming any type of beverages. At ~ 23:00, the participants completed a 10-minute PVT and their body water was measured using an InBody 770 (InBody Corp., Seoul, Korea) [29]. Bedtime was scheduled for 23:30. The experimental group consumed 280 mL of plain water, and the controls did not consume water or any other beverages immediately before bedtime. The amount of water was determined

| | | Adaptation night | Control or Water intake Condition | Washout period | | Adaptation night | Control or Water intake Condition |
|---|---|---|---|---|---|---|---|
| Days 1 - 4 | 5 | 6 - 7 | 8 - 14 | 15 - 18 | 19 | 20 - 21 | |

**Fig 2. Preparation before experiment.**

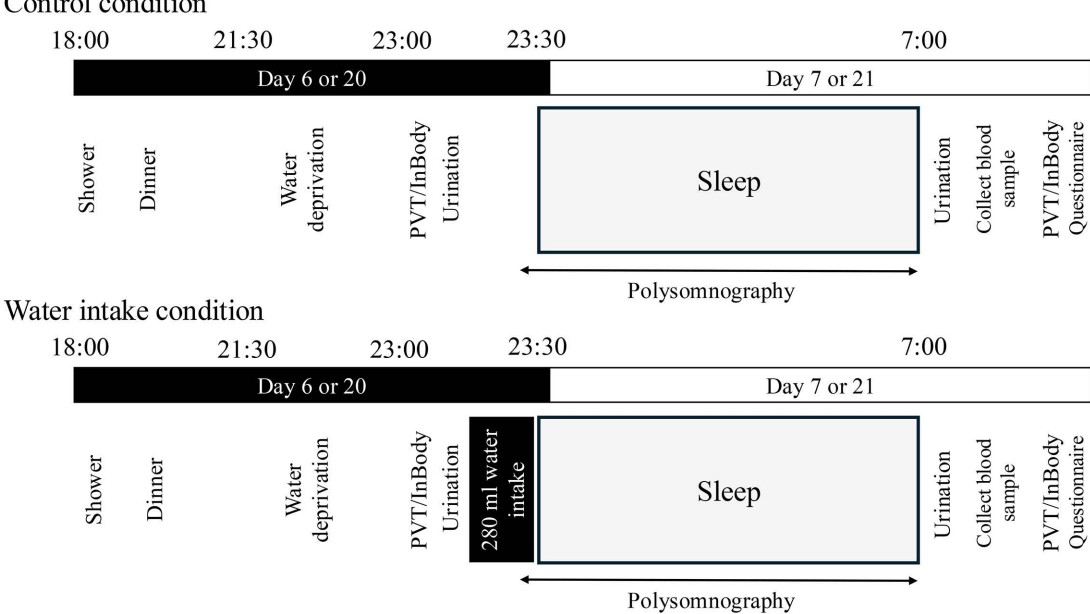

**Fig 3. Schedule of controls and water intake.**

based on calculations of average water loss for 2 hours before bedtime according to previous findings [30] with the intent being to avoid nocturia. An average water loss during sleep of 400–700 mL [31] was also considered. Around 07:30 h, the participants completed the questionnaires, then repeated the PVT and InBody measurements. The frequency of nocturia was determined as toilet flushing sounds during the night.

## Polysomnography (PSG)

Polysomnograms were assessed using an InSomnograf K2 portable multichannel electroencephalography recorder (S'UI-MIN Inc., Tokyo, Japan) [32] consisting of the electroencephalogram derivations, Fp1 - M2, Fp1 – average M, and Fp1 - Fp2 [33,34]. Professional staff manually scored sleep stages.

## Assessment of body water fluid status

We used an InBody 770 body composition and water analyzer that measures body composition and provides a detailed, complete breakdown of body water [29]. It analyzes bioelectrical analysis impedance (BIA) which measures the distribution of extracellular (ECW), intracellular (ICW) and total body water (TBW). The ratio of ECW to TBW (ECW/TBW) reflects the status of body water [35–37]. We used this ratio to validate the impact of water consumed before bedtime on the status of body water in the morning.

## Psychomotor vigilance test

Participants measured their performance and arousal level using a PVT-192 psychomotor vigilance monitor (Ambulatory Monitoring, Inc., Ardsley, NY, USA) for 10 min before and after bedtime. The PVT uses a simple visual reaction time (RT) paradigm with interstimulus intervals of 2,000–10,000 milliseconds [38]. Performance indexes (mean RTs, lapses > 500 ms RTs) were generated using PVTcmmW, v. 2.71/REACT, v. 1.1.03 (Ambulatory Monitoring Inc.). Mean RTs were transformed to reciprocal RTs (1/mean RTs) to avoid outliers.

## Questionnaires

The participants rated depressive moods using the CES-D scale and thirst upon waking using the thirst scale, as described in Study 1. They also scored nocturia using a 7-point Likert scale from "*none*" to "*extremely urgent.*"

## Compliance with ethical standards

The ethics review board at the National Institute of Advanced Industrial Science and Technology approved the study (Approval: ID 73002030-E-20240325–006), which also complied with the ethical principles enshrined in the Declaration of Helsinki (2013 amendment). Study 2 is listed in the UMIN Clinical Trials Registry (Approval: ID UMIN000054134, Registration Date: 05/01/2024). All respondents recruited between April 19th and July 10th 2024 provided written, informed consent to participate in this study.

## Statistical analysis

We analyzed the PVT and InBody results using repeated two-way ANOVA with time and condition followed by post-hoc Tukey HSD tests. Differences between conditions were assessed using paired $t$-tests. Values with $p < 0.05$ were considered statistically significant and $p < 0.10$ indicated marginally significant. All data were analyzed using IBM SPSS v. 29 for Windows.

## Results

**Water consumption one day before experimentation.** The volumes of diarized water intake calculated on the day of the experiment were 2.25 (SD = 0.75) and 2.32 (SD = 0.76) L in the control and experimental (water intake) groups, respectively ($t$ (54) = 1.37, $p = 0.18$, $d = 0.09$). The total amount of water intake by the experimental group was 2.60 L, including the 280 mL before bedtime ($t$ (54) = 6.82, $p < 0.01$, $d = 0.46$). The room temperature ($t$ (54) = 1.66, $p = 0.10$, $d = 0.18$) and humidity ($t$ (54) = 0.93, $p = 0.36$, $d = 0.12$) did not significantly differ throughout the study (control and experimental groups: 25.6°C ± 1.11°C and 70.7% ± 10.83% *vs.* 25.8°C ± 1.17°C and 69.5% ± 10.63%, respectively.

**Sleep parameters.** Table 1 shows the sleep parameters. Participants in the control and experimental groups respectively turned the room lights off at 23:48 (SD = 13.3) and 23:50 (SD = 12.3) min (t (54) = 0.72, $p = 0.48$, $d = 0.14$). The control and experimental groups slept from 00:07 (SD = 22.6 min) to 05:40 (SD = 34.1 min), and 00:07 (SD = 15.4 min) to 05:43 (SD = 32.2 min), respectively. Sleep latency ($t$ (54) = 0.03, $p = 0.97$, $d = 0.005$) and wakeup times ($t$ (54) = 0.84, $p = 0.41$, $d = 0.10$) did not significantly differ between the groups. The amount of REM sleep time tended to be shorter

**Table 1. Sleep parameters.**

| Parameters | Control (min ± SD) | Water intake (min ± SD) | t | p | Cohen's d |
|---|---|---|---|---|---|
| WASO | 40.9 ± 37.69 | 39.8 ± 25.57 | 0.27 | 0.79 | 0.04 |
| N1 | 35.5 ± 19.34 | 38.4 ± 20.25 | 1.28 | 0.21 | 0.17 |
| N2 | 164.7 ± 40.51 | 166.3 ± 34.54 | 0.32 | 0.75 | 0.04 |
| N3 (SWS) | 30.0 ± 27.04 | 32.4 ± 28.52 | 1.17 | 0.25 | 0.16 |
| REM | 69.4 ± 23.37 | 64.5 ± 23.13 | 1.91 | 0.06 | 0.26 |
| Sleep latency | 19.2 ± 21.71 | 17.4 ± 11.08 | 0.68 | 0.50 | 0.09 |
| REM latency | 77.7 ± 37.76 | 87.2 ± 36.79 | 2.38 | 0.02 | 0.32 |
| TST | 299.9 ± 51.82 | 303.4 ± 40.10 | 0.60 | 0.55 | 0.08 |

Abbreviations: min, minutes; N, non-REM; REM, rapid eye movement; SWS, slow wave sleep; TST, total sleep time. WASO, wake after sleep onset.

($t$ (54) = 1.91, $p < 0.10$, $d = 0.26$), and REM sleep latency was significantly longer ($t$ (54) = 2.38, $p < 0.05$, $d = 0.32$) in the experimental, than in the control group. Other sleep parameters did not significantly differ ($p > 0.10$).

**Body water status (ICW, ECW, TBW, ECW/TBW).** The ANOVA results show the main effect of water intake on ECW/TBW ($F$ (1, 54) = 6.55, $p < 0.01$, $\eta^2 = 0.11$) and time ($F$ (1, 54) = 207.1, $p < 0.01$, $\eta^2 = 0.79$). Interaction between them was not significant ($F$ (1, 54) = 0.10, $p = 0.76$, $\eta^2 = 0.002$). The main effect of time was significant for ICW, ECW, and TBW (all $p < 0.01$). These results indicated higher ECW/TBW ratios in the experimental, than in control group before and after sleeping (Fig 4). The ICW, ECW, TBW, and the ECW/TBW ratios decreased during sleep, and did not change due to water intake before bedtime (S2 and S3 Tables).

**Depression, thirst, and nocturia.** Depressive mood in the morning, measured by CES-D, tended to be lower in the experimental, than the control group ($t$ (54) = 1.18, $p < 0.10$, $d = 0.24$). The experimental group was not as thirsty as the control group ($t$ (54) = 1.76, $p < 0.10$, $d = 0.24$). Nocturia was more urgent in the experimental, than in the control group ($t$ (54) = 4.50, $p < 0.01$, $d = 0.61$). Fig 5 shows the responses to the questionnaire.

**Frequency of nocturia.** The frequency of nocturia represented as toilet flushing sounds, was calculated based on data from 54 participants, excluding one due to a technical error. The frequency of nocturia was significantly higher in the experimental, than in the control group (0.70 ± 0.439 vs. 0.24 ± 0.262; $t$ (53) = 5.93, $p < 0.01$, $d = 0.78$).

**PVT.** Table 2 shows that the PVT parameters did not significantly differ between the groups, and Table 3 shows the ANOVA results.

## Discussion

We investigated the effects of plain water intake before bedtime on sleep parameters and depressive mood. Study 1 revealed a negative correlation between pre-bedtime water intake and depressive mood in the following morning. Study 2 experimentally confirmed that consuming 280 mL of plain water immediately before bedtime improved morning mood.

Study 1 identified a negative relationship between water intake and depressive mood. Participants who consumed more water were less depressed. These results are consistent with previous findings indicating that water intake ameliorates depressive mood [7,11,12,17–21]. Insomnia scores were higher in groups 2–5 with moderate water intake than in the group 1 that not consumed water and in groups 6 and 7 that habitually consumed water. Group 1 did not intake water before bedtime and experienced the least thirst in the morning. This group might have already satisfied their hydration

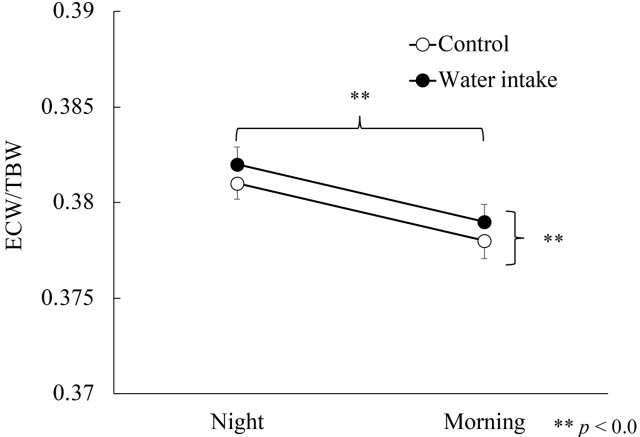

**Fig 4. Body water content before and after sleep.** The ECW/TBW ratio was higher at night than morning under both conditions. The ECW/TBW ratios were significantly elevated compared with the control group before and after sleep. ECW, extracellular water; TBW, total body water.

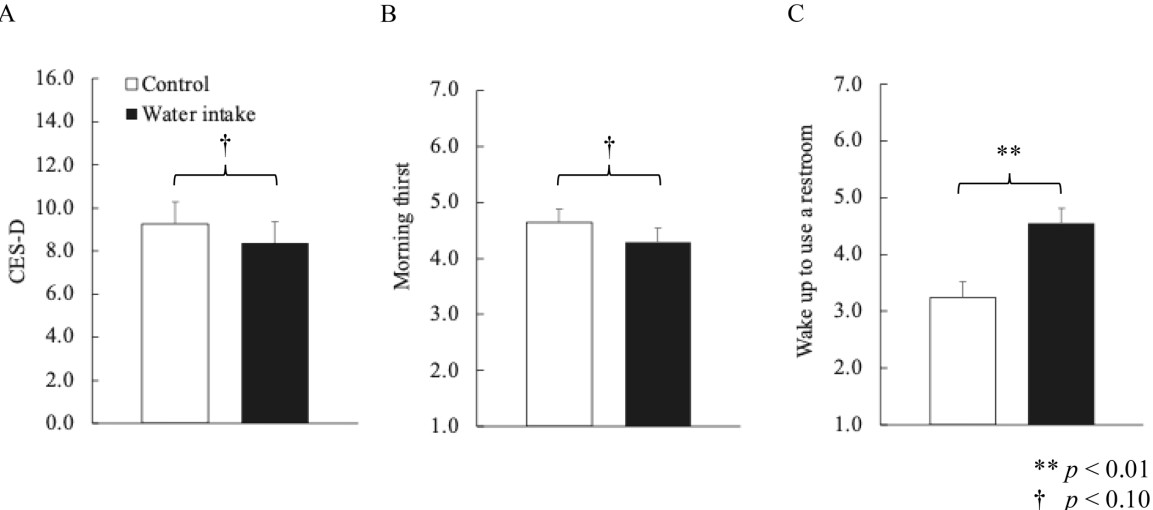

**\*\*** $p < 0.01$
**†** $p < 0.10$

**Fig 5. Questionnaire scores.** (A) Center for Epidemiologic Studies Depression Scale (CES-D) scores. Morning depressive mood tended to be lower in the experimental, than the control group. (B) Morning thirst was lower in the controls than the experimental group (C) Frequency of restroom visits caused by thirst was higher in the experimental group.

**Table 2. Results of psychomotor vigilance tests.**

| | Control | | Experimental | |
|---|---|---|---|---|
| | **Night** | **Morning** | **Night** | **Morning** |
| Reaction time (ms) | 247.2±31.46 | 251.5±42.53 | 244.3±29.92 | 246.3±27.79 |
| Lapse (frequency) | 0.9±1.13 | 1.2±2.49 | 1.0±1.46 | 0.7±1.39 |
| Error rate | 0.7±1.30 | 0.8±1.49 | 0.6±0.81 | 0.8±1.69 |

**Table 3. Results of psychomotor vigilance test (ANOVA).**

| | | df | *F* | *p* | η2 | Huynh-Feldt |
|---|---|---|---|---|---|---|
| Reaction time | | | | | | |
| | Time | 1, 54 | 1.12 | 0.30 | 0.02 | 1 |
| | Condition | 1, 54 | 1.37 | 0.25 | 0.02 | 1 |
| | Interaction | 1, 54 | 0.17 | 0.68 | 0.00 | 1 |
| Lapse | | | | | | |
| | Time | 1, 54 | 0.01 | 0.92 | 0.00 | 1 |
| | Condition | 1, 54 | 0.96 | 0.33 | 0.02 | 1 |
| | Interaction | 1, 54 | 1.91 | 0.17 | 0.03 | 1 |
| Error rate | | | | | | |
| | Time | 1, 54 | 0.68 | 0.41 | 0.01 | 1 |
| | Condition | 1, 54 | 0.29 | 0.59 | 0.01 | 1 |
| | Interaction | 1, 54 | 0.46 | 0.50 | 0.01 | 1 |

needs throughout the day. Groups 6 and 7 had lower insomnia scores than those with moderate water intake. Respondents who were thirsty in the morning tended to have insomnia. This suggests that water intake before bedtime contributes to euhydration during sleep and ameliorates insomnia and depressive mood.

Study 1 confirmed a positive correlation between thirst and depressive mood in the morning. A relationship between daytime feelings and thirst has been identified [17] and we revealed relationships between nighttime thirst and morning feelings. Our results were based on cross-sectional data, and therefore did not fully explain causality between water intake before bedtime and depressive mood upon waking. Therefore, we conducted an experimental study to clarify the causal relationship between water intake before bedtime and depressive mood.

Study 2: the duration of REM sleep was shorter, REM sleep latency was longer, and depressive mood in the morning tended to be lower in the experimental, than the control group. Although the neurophysiological mechanism is unknown, consuming 280 mL of plain water before bedtime ameliorated depressive mood with a REM sleep modification. A daytime causal relationship between plain water intake and depressive mood has been identified [7,11,12,17–21]. Here, we assessed the causal effect of water intake before bedtime and on REM sleep. Depressive mood might be associated with REM sleep latency and the amount of REM sleep [22–24,39]. We determined that plain water intake reduced the amount of REM sleep and increased REM sleep latency. Therefore, plain water intake might contribute to ameliorating depressive mood in the morning by modifying REM sleep. On the other hand, NREM sleep was not affected by plain water intake, which means that plain water intake might influence REM and NREM sleep *via* a different pathway.

Managing depressive mood through water intake might be an effective social solution in Japan and east Asia. The summer temperatures in most regions of Japan often exceed 30°C, with > 70% humidity (Japan Meteorological Agency; Overview of Japan's climate, 2024). Hydration under such conditions is crucial to prevent dehydration and heat-related issues, such as increased depressive mood [40]. The present study did not address treatment for clinical issues, but dehydration decreases blood volume, which can exacerbate cardiovascular strain and acute kidney failure [41]. Chronic dehydration can lead to kidney fibrosis and chronic kidney disease [42]. Drinking plain water before bedtime could be a simple and practical way to improve psychological and physical wellbeing, especially during the summer. Beverages containing electrolytes and/or carbohydrates are more effective than water for maintaining or recovering hydration [43,44]. The effects of such beverages on sleep and depressive mood could be a focus for future investigation.

Plain water intake prolonged REM sleep latency and reduced the amount of REM sleep. Although the mechanism remains obscure, we speculate that consuming water before bedtime stimulates gastrointestinal microbiome activity. Serotonergic activities in the gut are associated with the brain (microbiota-gut-brain axis) [45,46]. Serotonin ameliorates depression [47] and simultaneously reduces the amount of REM sleep [48]. The serotonergic system might explain the relationships among water intake, REM sleep, and depressive mood. However, the potentially complex mechanisms between water intake and REM sleep might be a worthwhile topic for future investigation.

Given the current findings on how water intake affects REM sleep and depressive mood, and the connection between REM sleep and psychiatric disorders, we propose that water intake helps to alleviate depressive mood by altering REM sleep profiles. However, the neuropsychiatric mechanism remains unclear [49]. Blood flow to the brain surface of mice increases during REM sleep, and this helps to expel neural waste that accumulates during waking hours [50]. Thus, refreshing neural waste during REM sleep and an enhanced mental state might be associated and if so, water intake before bedtime might support this process.

Plain water intake before sleep did not affect PVT performance after waking. Dehydration might [51,52] or might not [7,53–55] affect cognitive performance, but we found that mild hypohydration during sleep was unlikely to impact it. Extreme hypohydration (> 2% body weight) decreases cognitive performance [2,56,57]. Therefore, whether degrees of overnight hypohydration are associated with cognitive performance remains inconclusive. The insignificance of water intake on cognitive performance can also be explained based on our findings of its effects on depression. Cognitive responses are slower in patients with depression [58,59]. The present findings showed that plain water intake before sleep

reduced depressive mood in the morning, but did not apparently improve cognitive performance. This might be because our participants did not have notable depressive symptoms, reporting a normal range of CES-D scores mostly of 8–9. Thus, the effect of water intake on depression might not have been sufficient to improve PVT performance.

Plain water intake before bedtime reduced morning thirst and increased the frequency of nocturia. However, body water (ECW/TBW ratio) in the morning was not affected by water intake before bedtime. Despite the increase in nocturia after water intake, WASO did not always increase. This implies that the modest volume of water consumed before sleep did not significantly influence WASO, although the frequency of nocturia was slightly elevated in the experimental, than the control group the ($0.70 \pm 0.439$ *vs*. $0.24 \pm 0.262$). The increases in nocturia also indicated that most of the water consumed before bedtime was micturated, which would explain the increased urine volume during sleep. Although urine production to excrete water is suppressed at night *via* circadian rhythms [60], the results suggested that easing dehydration by consuming water before bedtime ameliorated depressive mood and insomnia, but simultaneously increased the frequency of nocturia, which is a negative side effect of hydration. Nocturia under dim light at night increases risk of falling, which is critical for elderly persons [61,62]. Having a bedroom close to a restroom can help lower risk of accidents involving nocturia, such as tripping or falling downstairs. The paradox of water intake before bedtime requires further exploration.

This study has a few limitations. The effect size was relatively small ($d = 0.20$–$0.50$, $\eta^2 = 0.01$–$0.06$), probably because physiological inter-individual differences such as sweat ability, kidney function, bladder size, surface skin area can be a latent confounding factor. Nonetheless, the present study revealed that plain water intake before sleep moderately improved mood status in the morning despite the small effect size. The CES-D scores of participants in Study 2 were ~8, which is much lower than the average Japanese CES-D score of ~12 [63]. Our relatively strict selection criteria for participation in the study might have resulted in a cohort of individuals with fairly good mental health status. This limitation might restrict the applicability of the current findings to a wider population that is generally mentally healthy. The results of Study 2 require validation based on average or severely depressed participants to provide reliable evidence to help people with depressive symptoms. The sleep parameters of WASO were relatively high (~40 min) and the amount of sleep was ~300 min compared to an average middle-aged (50s) Japanese population who slept for 6.5–6.8 hours and total WASO of 35–40 min [64]. The unfamiliarity of the hotel environment, combined with potential discomfort caused by the attached electrodes, might have disrupted the usual sleep behavior of the participants. Moreover, the experiment was conducted during the summer, which is characterized by seasonal variability in sleep duration [65], and record-breaking high temperatures in Japan in the summer of 2024 [66], both of which might have influenced the outcomes.

The present study notably found a higher proportion of body water (ECW/TBW) in the experimental, than the control group even before consuming 280 mL of water before bedtime. The intake of three meals and other foods before sleep was controlled throughout the study under both conditions, suggesting that these factors did not influence the difference in the ECW/TBW ratio. The difference might have resulted from consistent daily routines at home being changed by the presence or absence of water intake before bedtime, for six days before the experiment. Routinely drinking plain water before sleep might have altered ECW/TBW as a physiological state.

In conclusion, the present findings revealed that plain water intake at night before bedtime moderately reduced depressive mood in the morning, increased REM sleep latency, and reduced the duration of REM sleep. Of course, such intake increased the likelihood of nocturia. Despite the compromising advantages and disadvantages, plain water intake before bedtime is a simple and practical way to improve subjective wellbeing. Further studies including participants with depressive symptoms are warranted to generalize the findings.

## Supporting information

**S1 Table. Results of Tukey's HSD test (p values).**
(DOCX)

**S2 Table. Results of two-way repeated measures ANOVA for InBody measurements.**
(DOCX)

**S3 Table. Descriptive statistics of InBody measurements.**
(DOCX)

## Acknowledgments

We are grateful to Kiyoshi Yoshinaka, Naotaka Nitta, Mototada Shichiri, Hiroshi Endo, Sayaka Higo-Yamamoto, Chisa Emura for critical discussions. We thank CPCC Co., Ltd. (Tokyo, Japan) for conducting the clinical trial. We thank Hiroyuki Kato, Susumu Takemoto, Yasuhiro Tanaka for helpful comments regarding the experimental protocol and this draft of the manuscript.

## Author contributions

**Conceptualization:** Kosuke Kaida, Kazane Itoi, Yoshiharu Soeta, Toshihiko Ooie, Katsutaka Oishi.

**Data curation:** Kosuke Kaida, Kazane Itoi, Yoshiharu Soeta, Toshihiko Ooie, Katsutaka Oishi.

**Formal analysis:** Kosuke Kaida, Yoshiharu Soeta, Katsutaka Oishi.

**Funding acquisition:** Katsutaka Oishi.

**Investigation:** Kosuke Kaida, Kazane Itoi, Toshihiko Ooie, Katsutaka Oishi.

**Methodology:** Kosuke Kaida, Yoshiharu Soeta, Toshihiko Ooie, Katsutaka Oishi.

**Project administration:** Katsutaka Oishi.

**Supervision:** Katsutaka Oishi.

**Validation:** Kosuke Kaida, Kazane Itoi, Yoshiharu Soeta, Katsutaka Oishi.

**Visualization:** Kosuke Kaida, Kazane Itoi.

**Writing – original draft:** Kosuke Kaida, Kazane Itoi.

**Writing – review & editing:** Kosuke Kaida, Kazane Itoi, Katsutaka Oishi.

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
