## [Decision Letter · Decision Letter 0]

6 Aug 2025

Dear Dr. Kaida,

Thank you for submitting your manuscript to PLOS ONE. After careful consideration, we feel that it has merit but does not fully meet PLOS ONE’s publication criteria as it currently stands. Therefore, we invite you to submit a revised version of the manuscript that addresses the points raised during the review process.

We look forward to receiving your revised manuscript.

Kind regards,

Yosuke Yamada

Academic Editor

PLOS ONE

Journal Requirements:

We are grateful to Kiyoshi Yoshinaka, Naotaka Nitta, Mototada Shichiri, Hiroshi Endo, Sayaka Higo-Yamamoto, Chisa Emura for critical discussions. We thank CPCC Co., Ltd. (Tokyo, Japan) for conducting the clinical trial. We thank Suntory Beverage & Food Limited for financial support and Hiroyuki Kato, Susumu Takemoto, Yasuhiro Tanaka for helpful comments regarding the experimental protocol and this draft of the manuscript.

The study was funded by Suntory Beverage & Food Limited (Tokyo, Japan).

5. Please ensure that you refer to Figure 4 in your text as, if accepted, production will need this reference to link the reader to the figure.

Reviewers' comments:

Reviewer's Responses to Questions

**Comments to the Author**

1. Is the manuscript technically sound, and do the data support the conclusions?

Reviewer #1: Yes

Reviewer #2: No

2. Has the statistical analysis been performed appropriately and rigorously?

Reviewer #1: No

Reviewer #2: No

3. Have the authors made all data underlying the findings in their manuscript fully available?

Reviewer #1: Yes

Reviewer #2: No

4. Is the manuscript presented in an intelligible fashion and written in standard English?

Reviewer #1: No

Reviewer #2: Yes

Reviewer #1: This manuscript investigates the effects of plain water intake before bedtime on sleep parameters and depressive mood in middle-aged Japanese men. The study consists of two parts: a large-scale online survey and a laboratory-based experimental intervention. The research question is highly relevant and of significant public health interest, particularly given the potential for simple behavioral interventions to influence mental health and sleep quality.

However, there are several points where the manuscript could be improved to enhance clarity, scientific rigor, and overall readability.

1.There is a contradiction regarding REM sleep latency. The Abstract states that REM sleep latency was prolonged following water intake, whereas the General Discussion states that “plain water intake reduced REM sleep latency.” This is a critical inconsistency and should be corrected throughout the manuscript for clarity and accuracy.

2. The manuscript reports numerous t-tests without any mention of correction for multiple comparisons. Given the number of outcomes tested (e.g., sleep parameters, CES-D, PVT), the potential for Type I error is significant. Please discuss whether adjustments (e.g., Bonferroni correction, FDR) were considered.

Although Cohen’s d values are consistently reported, it would be helpful to interpret the effect sizes in context (e.g., small, medium, large).

3. While the negative correlation between water intake and CES-D scores in Study 1 is statistically significant (r = -0.18), the effect size is small. The authors should clarify the practical significance of this association and temper conclusions accordingly.

4. Study 2 participants reported CES-D scores averaging around 8, which is lower than the typical mean (~12) in the Japanese population. This suggests the sample was generally mentally healthy, which could limit the generalizability of findings. This limitation should be emphasized more explicitly in the Discussion.

5. Although the manuscript notes that nocturia frequency increased with pre-bedtime water intake, the clinical risks of nocturia, particularly among older adults (e.g., risk of falls and fractures), should be discussed in greater detail to provide context for the practical implications of this side effect.

6. Several sentences could be revised for improved academic English.

7. Some figure legends, such as Figure 5, are rather brief. Adding brief descriptions of the main findings shown in the figures would improve reader comprehension. Some abbreviations (e.g., ECW, TBW) should be defined in figure legends where first used.

Overall, this is a valuable and well-designed study addressing an underexplored topic of significant public health interest. The combination of observational and experimental approaches strengthens the validity of the findings. The required revisions relate mainly to clarifying inconsistencies, improving statistical reporting, and enhancing the clarity of language and interpretation. With these changes, the manuscript would be a strong candidate for publication in PLOS ONE.

Reviewer #2: General Comments

If this manuscript is intended to report two studies within a single paper, the current structure—where each study has separate background and discussion sections—makes the paper difficult to follow and disrupts its logical flow. I strongly recommend either:

1.Submitting the studies as two separate manuscripts, or 2.Presenting Study 1 and Study 2 only in the Methods and Results sections, while maintaining unified Introduction, Discussion, and Conclusion sections.

Additionally, the manuscript must conform to the structural format commonly accepted by the journal, namely: Introduction, Materials and Methods, Results, Discussion, and Conclusion. The current structure impairs readability and comprehension.

Major Comments

Page 14, Line 1 – Table 1

•The reported TST appears short (approximately 300 minutes). Was this due to the unfamiliar hotel environment? Although the Morningness-Eveningness Questionnaire indicated that participants were, on average, of intermediate chronotype, individual variation might have affected sleep duration under experimental conditions.

•Both trials show high WASO; could this also be due to discomfort in the hotel setting or increased nocturnal urination?

•Interestingly, despite the increase in nocturia, WASO did not increase accordingly. This may suggest a novel observation that warrants further discussion.

•Was wake-up time standardized? Did participants truly secure 7.5 hours of sleep as suggested in the figure?

Page 11, Line 4 – “average water loss during sleep (400–700 mL)”

•It is unclear how this estimate justifies the choice of 280 mL of plain water before bedtime. Please clarify the rationale behind selecting this amount.

Page 17, Line 19 – General Discussion

•The discussion focuses on concerns about dehydration, but it is unclear—both from the methods and the background—why plain water was selected as the rehydration fluid.

•If dehydration mitigation was a concern, previous studies have shown that beverages containing electrolytes and/or carbohydrates are more effective for hydration maintenance or recovery.

•The current data also suggest that plain water may lead to rapid excretion through urine, potentially increasing nocturnal urination and mid-sleep awakenings.

•However, WASO did not increase despite increased nocturia—please explain this apparent discrepancy.

Page 19, Line 2 – REM Latency and REM Sleep

•REM latency increased and REM duration decreased. Would such changes positively influence the clearance of neural waste products or psychological recovery during sleep?

•Line 57 seems to suggest the opposite. If I have misunderstood, I apologize—but clarification is needed as this appears contradictory.

Page 17, Line 26 – Statement: “We determined that plain water intake reduced the amount of REM sleep and increased REM sleep latency.”

•This conclusion seems inconsistent with the overall claims of the study. Please ensure alignment between the data and key conclusions.

Figure 4 – ECW/TBW Differences

•The difference in ECW/TBW ratios raises concern about experimental control. Although the study appears to have been carefully conducted, could the timing or content of pre-sleep meals or trial food have influenced these values?

•Please include discussion of potential confounding factors, including any interaction effects.

**Do you want your identity to be public for this peer review?** For information about this choice, including consent withdrawal, please see our Privacy Policy

Reviewer #1: No

Reviewer #2: No

---

## [Author Response · Author response to Decision Letter 1]

22 Sep 2025

Author’s responses to the editor and reviewers

We sincerely thank the academic editor Dr. Yosuke Yamada and anonymous experts for the time spent reviewing our manuscript and providing helpful comments. We have substantially revised our manuscript based on these comments and believe that it is much improved. Our point-by-point responses to each comment are listed below, and changes are shown in red in the revised manuscript. We hope that our manuscript will now be deemed suitable for publication in PLOS ONE.

Responses to Reviewer 1

This manuscript investigates the effects of plain water intake before bedtime on sleep parameters and depressive mood in middle-aged Japanese men. The study consists of two parts: a large-scale online survey and a laboratory-based experimental intervention. The research question is highly relevant and of significant public health interest, particularly given the potential for simple behavioral interventions to influence mental health and sleep quality. However, there are several points where the manuscript could be improved to enhance clarity, scientific rigor, and overall readability.

Comment 1: There is a contradiction regarding REM sleep latency. The Abstract states that REM sleep latency was prolonged following water intake, whereas the General Discussion states that “plain water intake reduced REM sleep latency.” This is a critical inconsistency and should be corrected throughout the manuscript for clarity and accuracy.

Response: We corrected the error in the Discussion section: “Plain water intake prolonged REM sleep latency and reduced the amount of REM sleep.” Page 17, line 12.

Comment 2: The manuscript reports numerous t-tests without any mention of correction for multiple comparisons. Given the number of outcomes tested (e.g., sleep parameters, CES-D, PVT), the potential for Type I error is significant. Please discuss whether adjustments (e.g., Bonferroni correction, FDR) were considered.

Response: We used only the t-test for one-time comparisons of independent data (Figure 5). Multiple comparisons (Figure 1) were assessed using Tukey HSD tests. We believe that Type I error could be avoided by applying the appropriate statistical analyses.

Comment 3: Although Cohen’s d values are consistently reported, it would be helpful to interpret the effect sizes in context (e.g., small, medium, large).

While the negative correlation between water intake and CES-D scores in Study 1 is statistically significant (r = -0.18), the effect size is small. The authors should clarify the practical significance of this association and temper conclusions accordingly.

Response: Small effect sizes (η2 = 0.01-0.06) can be explained to some extent by quite large substantial physiological differences such as sweating ability, kidney function, bladder size, and skin surface. Despite the small effect size, we believe that plain water intake is a relatively simple action with the potential to improve depressive mood in the morning. We weakened the conclusion and added a discussion about small effect sizes in the Discussion section. Page 19, line 5-9.

Comment 4: Study 2 participants reported CES-D scores averaging around 8, which is lower than the typical mean (~12) in the Japanese population. This suggests the sample was generally mentally healthy, which could limit the generalizability of findings. This limitation should be emphasized more explicitly in the Discussion.

Response: We added this as a limitation in the Discussion section: “Our relatively strict selection criteria for participating in the study might have resulted in a cohort of individuals with fairly good mental health status. This might restrict the applicability of our findings to a wider population that is generally mentally healthy.” Page 19, line 12-14.

Comment 5: Although the manuscript notes that nocturia frequency increased with pre-bedtime water intake, the clinical risks of nocturia, particularly among older adults (e.g., risk of falls and fractures), should be discussed in greater detail to provide context for the practical implications of this side effect.

Response: We added the practical implications of the side effect of nocturia. “Having a bedroom location close to a restroom can help lower accident risks involving nocturia, such as tripping on the floor or falling down the stairs. The paradox of water intake before bedtime requires further exploration.” Page 19, line 1-4.

Comment 6: Several sentences could be revised for improved academic English.

Response: A professional native English-speaking editor proof-checked our revised manuscript.

Comment 7

Some figure legends, such as Figure 5, are rather brief. Adding brief descriptions of the main findings shown in the figures would improve reader comprehension. Some abbreviations (e.g., ECW, TBW) should be defined in figure legends where first used.

Response: We added brief descriptions of the main findings to the figure legends and explained all abbreviations that appeared in figure legends. Page 20, Legends.

Comment 8: Overall, this is a valuable and well-designed study addressing an underexplored topic of significant public health interest. The combination of observational and experimental approaches strengthens the validity of the findings. The required revisions relate mainly to clarifying inconsistencies, improving statistical reporting, and enhancing the clarity of language and interpretation. With these changes, the manuscript would be a strong candidate for publication in PLOS ONE.

Response: We are very much encouraged by this comment.

Comments from Reviewer 2

Comment 9: If this manuscript is intended to report two studies within a single paper, the current structure—where each study has separate background and discussion sections—makes the paper difficult to follow and disrupts its logical flow. I strongly recommend either:

1.Submitting the studies as two separate manuscripts, or 2.Presenting Study 1 and Study 2 only in the Methods and Results sections, while maintaining unified Introduction, Discussion, and Conclusion sections.

Additionally, the manuscript must conform to the structural format commonly accepted by the journal, namely: Introduction, Materials and Methods, Results, Discussion, and Conclusion. The current structure impairs readability and comprehension.

Response: We merged the Introduction and Discussion sections of the two studies.

Comment 10-1

Page 14, Line 1 – Table 1

•The reported TST appears short (approximately 300 minutes). Was this due to the unfamiliar hotel environment? Although the Morningness-Eveningness Questionnaire indicated that participants were, on average, of intermediate chronotype, individual variation might have affected sleep duration under experimental conditions.

•Both trials show high WASO; could this also be due to discomfort in the hotel setting or increased nocturnal urination?

Response: The reasons for the reduced total amount of sleep in our participants compared to the average reported for Japanese individuals in their 50s (~ 6.34 hours), and high WASO remains unclear. The study was a within-subject design with a counterbalanced condition order, thereby minimizing potential inter-individual variability. Nevertheless, the experiment proceeded during the summer, when seasonal fluctuations in sleep duration (Suzuki et al., 2019, Plos One, e0215345) and the unprecedented high temperatures in Japan recorded during the summer of 2024 (Takemura et al., 2025, SOLA, 329–339) might have been factors. Furthermore, sleeping in a hotel might also have indeed impacted sleep duration. We added this as a limitation in the Discussion section. Page 19, line 16-23.

Comment 10-2

•Interestingly, despite the increase in nocturia, WASO did not increase accordingly. This may suggest a novel observation that warrants further discussion.

•The current data also suggest that plain water may lead to rapid excretion through urine, potentially increasing nocturnal urination and mid-sleep awakenings.

•However, WASO did not increase despite increased nocturia—please explain this apparent discrepancy.

Response: We believe that the time taken to use the bathroom did not significantly affect WASO. The discrepancy between the increase in nocturia but not in WASO can be explained by the fact that the difference in frequent nocturia was not so large between the experimental and control conditions (0.70 ± 0.439 vs. 0.24 ± 0.262, respectively). The small amount (280 mL) of water intake before sleep might not be sufficient to affect WASO, even though it slightly increased the frequency of nocturia. We added this point to the Discussion section. Page 18, line 17-21.

Comment 10-3

•Was wake-up time standardized? Did participants truly secure 7.5 hours of sleep as suggested in the figure?

Response: Figure 3 shows the standard schedule planned in the experiment protocol, and it does not mean that the participants had 7.5 hours of sleep. The total sleep time was around 300 min and differed among participants within the ranges shown in Table 1. The time was measured by polysomnography.

Comment 11: Page 11, Line 4 – “average water loss during sleep (400–700 mL)”

•It is unclear how this estimate justifies the choice of 280 mL of plain water before bedtime. Please clarify the rationale behind selecting this amount.

Response: The reported average water loss during sleep is 400–700 mL, which could cause nocturia. Considering nocturia, we set the water intake amount to 280 mL based on the average amount of water lost during 2 hours before bedtime (Yamada et al., 2022). We explained the rationale in the Methods section. Page 10, line 4-6.

Comment 12: Page 17, Line 19 – General Discussion

•The discussion focuses on concerns about dehydration, but it is unclear—both from the methods and the background—why plain water was selected as the rehydration fluid.

•If dehydration mitigation was a concern, previous studies have shown that beverages containing electrolytes and/or carbohydrates are more effective for hydration maintenance or recovery.

Response: Indeed beverages containing electrolytes and/or carbohydrates are more effective for maintaining or recovering hydration than plain water. We aimed to obtain fundamental information about the impact of rehydration immediately before sleep on depressive mood in older adult men after waking up in the morning. This is first investigation of this topic as far as we can ascertain. We therefore tested our hypotheses using plain water because it is the most basic beverage consumed by humans for rehydration compared with beverages containing electrolytes and/or carbohydrates. This matter is described in the Discussion section. Page 17, line 8-11.

Comment 13-1

Page 19, Line 2 – REM Latency and REM Sleep

•REM latency increased and REM duration decreased. Would such changes positively influence the clearance of neural waste products or psychological recovery during sleep?

Response: A study of mice (Tsai et al., 2021) found that REM sleep might be associated with the clearance of neural waste substances, but this has not been proven in humans. Therefore, we mentioned this as a possible cause of reduced depressive mood in the participants who consumed water compared to those who did not.

Comment 13-2

•Line 57 seems to suggest the opposite. If I have misunderstood, I apologize—but clarification is needed as this appears contradictory.

Page 17, Line 26 – Statement: “We determined that plain water intake reduced the amount of REM sleep and increased REM sleep latency.”

•This conclusion seems inconsistent with the overall claims of the study. Please ensure alignment between the data and key conclusions.

Response: We corrected this error.

Comment 14: Figure 4 – ECW/TBW Differences

•The difference in ECW/TBW ratios raises concern about experimental control. Although the study appears to have been carefully conducted, could the timing or content of pre-sleep meals or trial food have influenced these values?

Response: During the experiment, three meals and anything else consumed before sleeping were controlled under both conditions, so we believe that they did not affect the difference in ECW/TBW. We assume that the difference in ECW/TBW before sleep between the conditions was due to habituation from consecutive nights of water intake over a week. We added this as a limitation to the Discussion section.

Comment 15-2

•Please include discussion of potential confounding factors, including any interaction effects.

Response: Physiological inter-individual differences such as sweating ability, kidney function, bladder size, and surface skin area could be latent confounding factors. This is mentioned in the Discussion section.

Journal Requirements

Response: We changed the file names to meet the style requirements of PLOS ONE.

Response: We corrected the mismatch between “Funding Information” and “Financial Disclosure” removed funding-related text from the manuscript, and included the amended statements in our cover letter.

We are grateful to Kiyoshi Yoshinaka, Naotaka Nitta, Mototada Shichiri, Hiroshi Endo, Sayaka Higo-Yamamoto, Chisa Emura for critical discussions. We thank CPCC Co., Ltd. (Tokyo, Japan) for conducting the clinical trial. We thank Suntory Beverage & Food Limited for financial support and Hiroyuki Kato, Susumu Takemoto, Yasuhiro Tanaka for helpful comments regarding the experimental protocol and this draft of the manuscript.

The study was funded by Suntory Beverage & Food Limited (Tokyo, Japan).

Response: We removed funding-related text from the manuscript and included the amended statements within our cover letter.

b) If there are no restrictions, please upload the minimal anonymized data set necessary to replicate your study findings to a stable, public repository and provide us with the relevant URLs, DOIs, or accession numbers. Please see http://www.bmj.com/content/340/bmj.c181.long for guidelines on how to de-identify and prepar

---

## [Decision Letter · Decision Letter 1]

11 Dec 2025

Dear Dr. Kaida,

Thank you for submitting your manuscript to PLOS ONE. After careful consideration, we feel that it has merit but does not fully meet PLOS ONE’s publication criteria as it currently stands. Therefore, we invite you to submit a revised version of the manuscript that addresses the points raised during the review process. Please submit your revised manuscript by Jan 25 2026 11:59PM. If you will need more time than this to complete your revisions, please reply to this message or contact the journal office at plosone@plos.org . A rebuttal letter that responds to each point raised by the academic editor and reviewer(s). You should upload this letter as a separate file labeled 'Response to Reviewers'.A marked-up copy of your manuscript that highlights changes made to the original version. You should upload this as a separate file labeled 'Revised Manuscript with Track Changes'.An unmarked version of your revised paper without tracked changes. You should upload this as a separate file labeled 'Manuscript'.

We look forward to receiving your revised manuscript.

Kind regards,

Yosuke Yamada

Academic Editor

PLOS One

Journal Requirements:

Reviewers' comments:

Reviewer's Responses to Questions

**Comments to the Author**

Reviewer #1: All comments have been addressed

Reviewer #2: (No Response)

2. Is the manuscript technically sound, and do the data support the conclusions?

Reviewer #1: Yes

Reviewer #2: Yes

3. Has the statistical analysis been performed appropriately and rigorously?

Reviewer #1: Yes

Reviewer #2: Yes

4. Have the authors made all data underlying the findings in their manuscript fully available?

Reviewer #1: Yes

Reviewer #2: Yes

5. Is the manuscript presented in an intelligible fashion and written in standard English?

Reviewer #1: Yes

Reviewer #2: Yes

Reviewer #1: The authors have addressed all prior reviewer and editorial concerns thoroughly.

Major issues such as (1) inconsistency in REM sleep latency, (2) statistical transparency, (3) data availability, and (4) funding statement alignment have been satisfactorily resolved. The revised version demonstrates careful editing and a clear methodological rationale.

The manuscript now meets the criteria for publication in PLOS ONE. The scientific quality is sound, and the conclusions are appropriately cautious and supported by the data. Minor textual polishing may still be needed (e.g., standardizing expressions of “trend” to “marginally significant”), but these do not affect the overall integrity of the work.

This revised version reflects substantial improvement. The writing is clear, the structure coherent, and the key revisions have resolved the previous methodological and interpretive concerns. I appreciate the authors’ careful attention to detail and transparent responses to every comment.

* The contradiction regarding REM sleep latency and duration has been corrected and now aligns with the reported findings.

* The Discussion appropriately acknowledges confounding factors (hydration status, ECW/TBW differences, seasonal and environmental effects).

* Statistical methods and figure legends have been clarified; multiple comparison procedures are now explicitly described.

* Funding, ethics, and data availability statements fully comply with PLOS ONE requirements.

Minor Points for Further Improvement

1. Statistical phrasing: Replace “trend” with “marginally significant (p = …)” for clarity and to conform to journal style.

2. Conclusion tone: The last paragraph (“advantages and disadvantages”) could end with a more forward-looking statement, such as: Further studies including participants with depressive symptoms are warranted to generalize the findings.

3. Language consistency:Consider minor stylistic edits (e.g.,ameliorated depressive mood → “improved morning mood”) for smoother readability.

4. Data statement: Confirm that Supporting Information files correspond exactly to the datasets referenced in the Results.

Overall, this version is scientifically sound and clearly presented. I commend the authors for their comprehensive and thoughtful revision.

Reviewer #2: I have no further major comments. However, I noticed that the number of subjects in the manuscript is slightly different from the UMIN registration (UMIN000054134, Registration Date: 05/01/2024). May be 62.

**Do you want your identity to be public for this peer review?** For information about this choice, including consent withdrawal, please see our Privacy Policy

Reviewer #1: No

Reviewer #2: No

---

## [Author Response · Author response to Decision Letter 2]

17 Dec 2025

Author’s responses to the editor and reviewers

We sincerely reiterate our gratitude to the Academic Editor, Dr. Yosuke Yamada, and the anonymous reviewers for the time and effort devoted to the evaluation of our manuscript and for their constructive and insightful comments. Our detailed, point-by-point responses to each remark are presented below, and all revisions have been highlighted in red in the revised manuscript. We believe that, in its current form, the manuscript will now be considered suitable for publication in PLOS ONE.

Responses to Reviewer 1

The authors have addressed all prior reviewer and editorial concerns thoroughly.

Major issues such as (1) inconsistency in REM sleep latency, (2) statistical transparency, (3) data availability, and (4) funding statement alignment have been satisfactorily resolved. The revised version demonstrates careful editing and a clear methodological rationale.

The manuscript now meets the criteria for publication in PLOS ONE. The scientific quality is sound, and the conclusions are appropriately cautious and supported by the data. Minor textual polishing may still be needed (e.g., standardizing expressions of “trend” to “marginally significant”), but these do not affect the overall integrity of the work.

This revised version reflects substantial improvement. The writing is clear, the structure coherent, and the key revisions have resolved the previous methodological and interpretive concerns. I appreciate the authors’ careful attention to detail and transparent responses to every comment.

* The contradiction regarding REM sleep latency and duration has been corrected and now aligns with the reported findings.

* The Discussion appropriately acknowledges confounding factors (hydration status, ECW/TBW differences, seasonal and environmental effects).

* Statistical methods and figure legends have been clarified; multiple comparison procedures are now explicitly described.

* Funding, ethics, and data availability statements fully comply with PLOS ONE requirements.

Comment 1:

Statistical phrasing: Replace “trend” with “marginally significant (p = …)” for clarity and to conform to journal style.

Response:

We employed the term “marginally significant” to describe the statistical outcome. See page 11, line 26.

Comment 2:

Conclusion tone: The last paragraph (“advantages and disadvantages”) could end with a more forward-looking statement, such as: Further studies including participants with depressive symptoms are warranted to generalize the findings.

Response:

We appended the statement, “Further studies including participants with depressive symptoms are warranted to generalize the findings.” to the concluding sentence. Thank you for your insightful suggestion. See page 20, line 11-12.

Comment 3:

Language consistency: Consider minor stylistic edits (e.g., ameliorated depressive mood → “improved morning mood”) for smoother readability.

Response:

We revised the phrase “ameliorated depressive mood in the morning” to “improved morning mood” to enhance readability. See page 15, line 8-9.

Comment 4:

Data statement: Confirm that Supporting Information files correspond exactly to the datasets referenced in the Results.

Response:

We reviewed the Supplement Tables and revised the table nomenclature presented in the Results section. See page 13, line 11.

Comment 5:

Overall, this version is scientifically sound and clearly presented. I commend the authors for their comprehensive and thoughtful revision.

Response:

Thank you very much for your constructive comments, which have strengthened our manuscript.

Responses to Reviewer 2

Comment 1:

I have no further major comments. However, I noticed that the number of subjects in the manuscript is slightly different from the UMIN registration (UMIN000054134, Registration Date: 05/01/2024). May be 62.

Response:

The initial planning phase involved 60 participants. Owing to attrition during the trial, the cohort was expanded to 63 individuals. At the data analysis stage, missing data reduced the final sample size to 55 participants. Thank you for your comment.

---

## [Editor Report · Decision Letter 2]

22 Dec 2025

Effects of plain water intake before bedtime on sleep and depressive mood among middle-aged Japanese men.

PONE-D-25-22869R2

Dear Dr. Kaida,

We’re pleased to inform you that your manuscript has been judged scientifically suitable for publication and will be formally accepted for publication once it meets all outstanding technical requirements.

Kind regards,

Yosuke Yamada

Academic Editor

PLOS One
---

## [Editor Report · Acceptance letter]

PONE-D-25-22869R2

PLOS One

Dear Dr. Kaida,

I'm pleased to inform you that your manuscript has been deemed suitable for publication in PLOS One. Congratulations! Your manuscript is now being handed over to our production team.

Kind regards,

on behalf of

Dr. Yosuke Yamada

Academic Editor

PLOS One